# Overlapping Phenotype of Cardiomyopathy in a Patient with Double Mutation: A Case Report

**Sigita Glaveckaitė** [1,*] , **Violeta Mikštienė** [2] , **Eglė Preikšaitienė** [3] , **Rimvydas Norvilas** [3] , **Ramūnas Janavičius** [2,3] **and Nomeda Rima Valevičienė** [4]

1 Department of Cardiovascular Diseases, Faculty of Medicine, Institute of Clinical Medicine, Vilnius University, 03101 Vilnius, Lithuania
2 Department of Human and Medical Genetics, Faculty of Medicine, Institute of Biomedical Sciences, Vilnius University, 03101 Vilnius, Lithuania; violeta.mikstiene@santa.lt (V.M.); ramunas.janavicius@santa.lt (R.J.)
3 Hematology, Oncology and Transfusion Medicine Center, Vilnius University Hospital Santaros Klinikos, 08661 Vilnius, Lithuania; egle.preiksaitiene@santa.lt (E.P.); rimvydas.norvilas@santa.lt (R.N.)
4 Department of Radiology, Nuclear Medicine and Medical Physics, Faculty of Medicine, Institute of Biomedical Sciences, Vilnius University, 03101 Vilnius, Lithuania; nomeda.valeviciene@santa.lt
* Correspondence: sigita.glaveckaite@gmail.com; Tel.: +37-068-240-937

**Abstract:** Hypertrophic cardiomyopathy and left ventricular noncompaction commonly occur as separate disorders with distinct clinical and pathoanatomical features. However, these cardiomyopathies may have a similar genetic origin with mutations encoding sarcomeric proteins. The described case report demonstrates an example in which phenotypic expression of both diseases occurred in the same patient, who has two different alterations; one of them is a likely pathogenic variant in the MYL3 gene (MIM#160790) and the second variant in the MYH6 gene (MIM#160710) of unknown significance so far. To better understand associations between specific genetic variants and phenotypical expression of these genetic alterations and to stratify patient risk and decide on the most appropriate treatment, a comprehensive multimodality imaging approach and experienced multidisciplinary cardiomyopathy team decisions are warranted. In the clinical routine, awareness of the existence of complex cardiomyopathy phenotypes should be paid more attention during echocardiographic examination and should encourage a broader use of cardiovascular magnetic resonance.

**Keywords:** left ventricular noncompaction; apical hypertrophic cardiomyopathy; next-generation sequencing; case report



## 1. Introduction

Hereditary cardiomyopathies (CMs) represent a very large and heterogeneous group of inherited heart disorders. According to the results of instrumental evaluation, many types of the disease have been characterized: hypertrophic CM (HCM), left ventricular noncompaction (LVNC), dilated CM, and arrhythmogenic right ventricle CM to mention the most frequent entities. LVNC (ORPHA:54260) and HCM (ORPHA:217569) commonly occur as separate disorders with distinct clinical and pathoanatomical features [1,2]. However, in some patients, overlapping or mixed phenotypes are diagnosed only based on the use of sophisticated imaging modalities, especially cardiovascular magnetic resonance (CMR). Unfortunately, sometimes the phenotypic manifestation of overlapping phenotypes poses some difficulties in determination of the particular disorder which is essential in the treatment and surveillance of the patient. Moreover, the same genes may be implicated in the pathogenesis of different CMs, making diagnostics even more complicated. Pathogenic variants in MYBPC3 and MYH7 genes are responsible for the development of most non-syndromic HCMs and a big part of LVNC cases, although more than 30 genes are currently associated with the pathogenesis of these disorders.

In this case report, the authors present two patients from one nuclear family suffering with hereditary heart disorders possessing variants in the MYL3 gene (MIM#160790) and the MYH6 gene (MIM#160710). Although we know that the pathogenic variants of the MYL3 gene have been associated with familial HCM, the alteration in the MYH6 gene NM_002471.3:c.169G>A, NP_002462.2:p.(Gly57Ser) has not been described in the scientific literature previously. The genetic change was classified as variant of unknown significance (VUS) according to the American College of Medical Genetics and Genomics (ACMG) criteria. Based on the case presented, we postulate that the presence of this double mutation causes an overlapping hypertrophic–noncompaction phenotype. Additionally, we emphasize the utility of multimodality imaging for the phenotypical assessment of cardiomyopathy patients, as well as a cardiomyopathy team approach to choose the most suitable treatment.

## 2. Methods

Next-generation sequencing analysis of genomic DNA isolated from two patients' peripheral blood was performed using TruSight Cardio Sequencing panel (Illumina Inc., San Diego, CA, USA). A total of 174 genes (coding exons) were analyzed, including the main genes associated with cardiomyopathies (*ABCC9*, *ACTC1*, *ACTN2*, *ANKRD1*, *BRAF*, *CAV3*, *CBL*, *CRYAB*, *CSRP3*, *DES*, *DSC2*, *DSG2*, *DSP*, *DTNA*, *GAA*, *GLA*, *HFE*, *HRAS*, *JUP*, *KRAS*, *LAMA4*, *LAMP2*, *LDB3*, *LMNA*, *MAP2K1*, *MAP2K2*, *MYBPC3*, *MYH6*, *MYH7*, *MYL2*, *MYL3*, *MYLK2*, *MYOZ2*, *MYPN*, *NEXN*, *NRAS*, *PKP2*, *PLN*, *PRDM16*, *PRKAG2*, *PTPN11*, *RAF1*, *RBM20*, *RYR2*, *SCN5A*, *SGCD*, *SHOC2*, *SOS1*, *TAZ*, *TCAP*, *TGFB3*, *TMEM43*, *TNNC1*, *TNNI3*, *TNNT2*, *TPM1*, *TTN*, *TTR*, *VCL*). Prepared DNA libraries were sequenced on the Illumina MiSeq system. The combined coverage was 572 kbp in sequence length. Data analysis was performed using standard Illumina bioinformatic workflow. Detected gene variants were analyzed and annotated using the VariantStudio 3.0 software. Synonymous or intronic variants and variants with a minor allele frequency of less than 2% were excluded. In silico analysis of missense mutations was performed using PolyPhen-2 (http://genetics.bwh.harvard.edu/pph2/ (accessed on 4 August2019)), SIFT Human Protein (http://sift.jcvi.org/ (accessed on 4 August2019)), and Mutation Taster (www.mutationtaster.org/ (accessed on 4 August2019)).

Polymerase chain reactions (PCR) of gDNA sequences flanking variant NM_000258.2: c.382G>T, NP_000249.1:p.(Gly128Cys), rs199474704 of the *MYL3* gene and variant NM_002 471.3:c.169G>A, NP_002462.2:p.(Gly57Ser) of the *MYH6* gene were performed using specific primers designed with the Primer Blast tool [3,4]. The PCR products were sequenced using the BigDye® Terminator v3.1 Cycle Sequencing Kit (Thermo Fisher Scientific, Waldham, MA, USA) and the ABI 3130xL Genetic Analyzer (Thermo Fisher Scientific, Waldham, MA, USA). The sequences were aligned with the reference sequence of the *MYL3* (NCBI: NM_000258.2) and *MYH6* (NCBI: NM_002471.3) genes.

Two-dimensional transthoracic echocardiography (TTE) using an ultrasonic system equipped with 1.5–4.5 MHz transducer (GE Vivid E9, GE Healthcare, New York, NY, USA) and 1.5 T cardiovascular magnetic resonance (CMR) (Siemens Avanto, Erlangen, Germany) was used to assess the specific CM phenotype.

## 3. Case Report

A 39-year-old male consulted a cardiologist on an outpatient basis due to nonanginal chest pain episodes five years ago and was diagnosed with isolated apical HCM by using TTE. At the time of follow-up consultation, the patient was asymptomatic with an unremarkable personal history and a family history of unspecified congenital heart disease and sudden cardiac death of his sister and his aunt. His physical examination was without any pathological findings. Electrocardiogram showed sinus rhythm, 80 bpm, hypertrophy of left ventricle (LV), and deep negative T-waves in leads I, aVL, V4–6. Blood biochemistry was normal. TTE was performed and demonstrated isolated apical HCM with apical micro-aneurysm. Interestingly, blood flow was registered in the projection of

hypertrophied midventricular segments, raising the question about the correctness of the previous diagnosis. CMR was scheduled for clarification of the heart's morphology and function, as well as for additional risk stratification of the patient. The CMR revealed hyperkinetic LV with a hyperkinetic ejection fraction around 84% and overlapping phenotypical pattern of LV myocardium with hypertrophy of compacted apical segments (maximum wall thickness up to 16 mm (Figure 1E,F) and hypertrabeculation of midventricular segments with a ratio of non-compacted and compacted myocardium up to 2.8 at end-diastole (Figure 1C,D; Supplementary Videos S1–S5). Additionally, LV apical micro-aneurysm (Figure 1A,B) was detected with transmural late gadolinium enhancement (LGE) in its wall, showing transmural fibrotic changes (Figure 2A). To exclude any coronary artery anomalies or underlying coronary artery disease coronary, computed tomography (CT) angiography was performed which demonstrated normal coronary arteries without any atherosclerotic changes or anomalies with morphological changes of LV consistent with the CMR findings (Figure 2B). Then, 24 h ECG monitoring was performed and revealed four sporadic ventricular premature beats.

The patient underwent genetic consultation and testing. Phenotypic evaluation revealed a non-syndromic type of HCM. Genealogy analysis showed multiple individuals affected with cardiac disorder in the family (Supplementary Figure S1). The father died of myocardial infarction at the age of 72 years; the mother suffered from heart rhythm disorder and died at the age of 64 years. The sister of the patient experienced sudden cardiac death at the age of 5 years, having an unspecified "congenital" heart defect. The genetic testing of the patient revealed a heterozygous missense type *MYL3* gene variant NM_000258.2c.382G>T, NP_000249.1:p.(Gly128Cys), rs199474704 and a heterozygous missense type *MYH6* gene variant NM_002471.3:c.169G>A, NP_002462.2:p.(Gly57Ser).

Additionally, the patient's 15-year-old daughter was invited for cardiological examination and genetic consultation and testing. CMR revealed normal LV systolic function without evidence of LV hypertrophy. However, hypertrabeculation of the apical to midventricular segments was observed, with a ratio of non-compacted to compacted myocardium up to 2.0, which was not diagnostic for left ventricular noncompaction (see Supplementary Videos 6–7). Subsequently, 24 h ECG monitoring was performed and revealed one sporadic supraventricular premature beat. Sanger sequencing confirmed that the patient had passed both *MYL3* gene variant c.382G>T and *MYH6* gene variant c.169G>A to his affected daughter (Supplementary Figures S2 and S3).

The above-described genetic alterations were discussed among a dedicated cardiomyopathy team, taking into account the presence of double genetic variants and LV apical micro-aneurysm with fibrotic changes and considering the patient's preferences. The decision was to schedule the patient for implantation of a cardioverter-defibrillator (ICD) for primary prevention of sudden cardiac death. Treatment with beta-blockers (47.5 mg of metoprolol succinate) was prescribed, and the patient was referred for regular cardiological follow-up. Two years following, the patient remains asymptomatic without any events registered by the ICD events so far. The patient's daughter was scheduled for a cardiological follow-up.

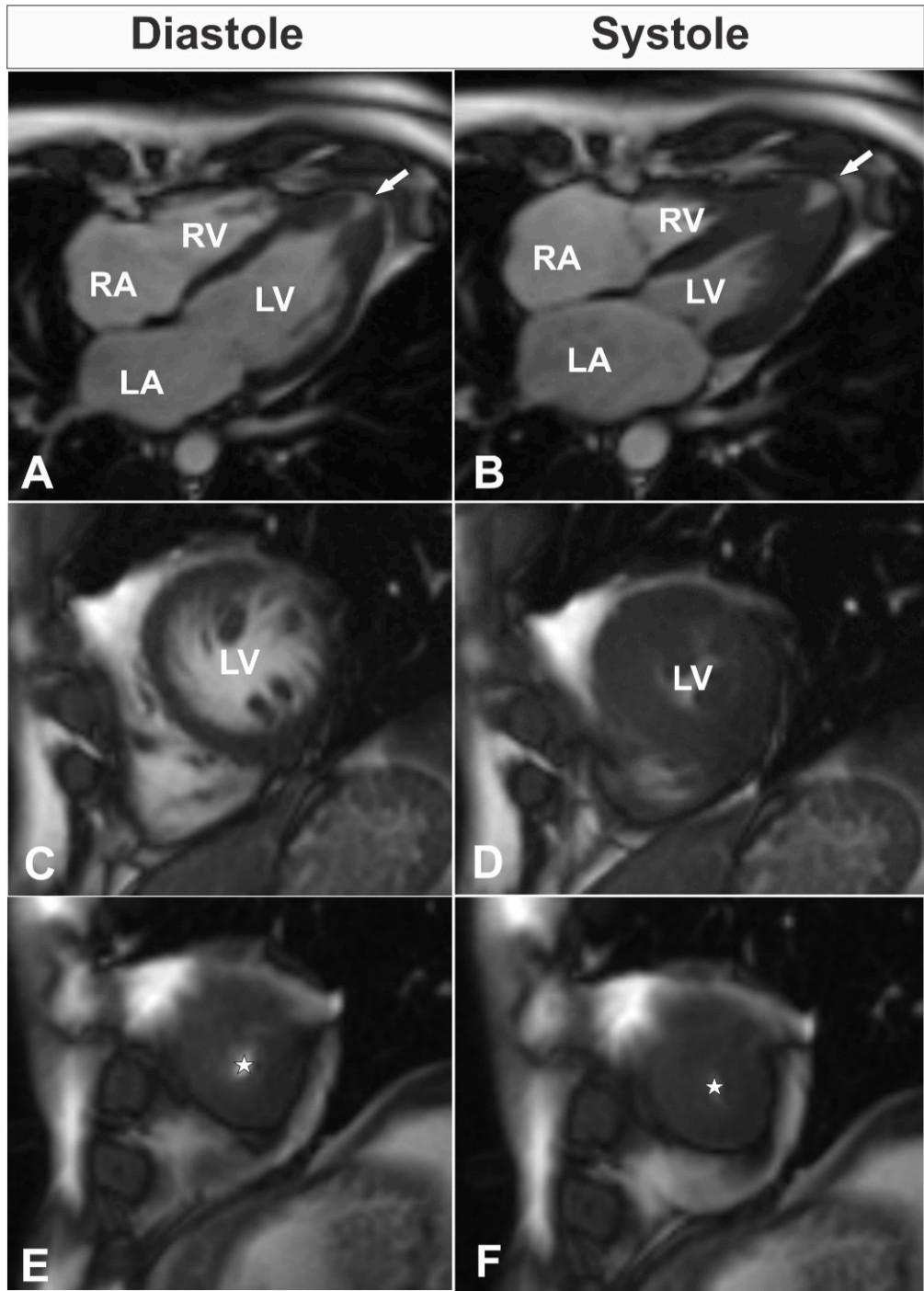

**Figure 1.** Cine magnetic resonance (steady state free precession sequence) heart views in diastole (left-sided column) and systole (right-sided column). (**A,B**) Four-chamber cine views; (**C,D**) midventricular short axis cine views; (**E,F**) apical short axis cine views. LA—left atrium; LV—left ventricle; RA—right atrium; RV—right ventricle. * Blood pool.

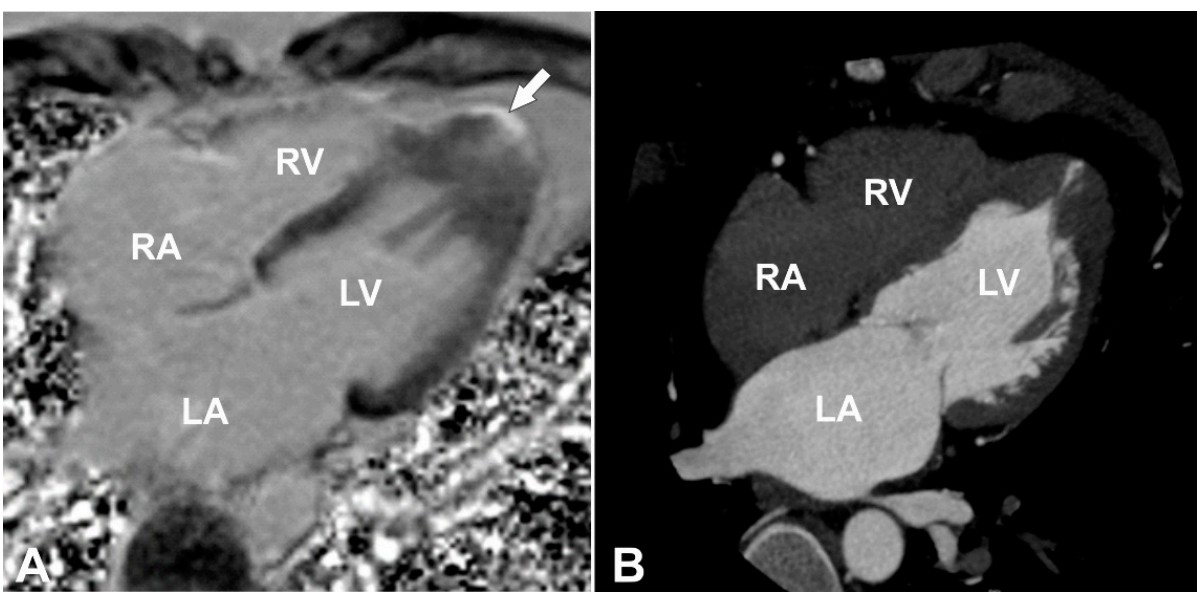

**Figure 2.** (**A**) Magnetic resonance four chamber heart view obtained with inversion recovery sequence 10 min after injection of gadolinium-based contrast agent (late gadolinium enhancement (LGE) image) with apical fibrotic changes denoted with a white arrow. (**B**) Four-chamber computed tomography heart view representing the morphologic findings of the heart seen on magnetic resonance images. LA—left atrium; LV—left ventricle; RA—right atrium; RV—right ventricle.

## 4. Discussion

With this case report, the authors present two patients from one nuclear family suffering from hereditary heart disorders possessing variants in the *MYL3* gene (MIM#160790) and the *MYH6* gene (MIM#160710). Based on the case description, we postulate that the presence of the particular double mutation causes an overlapping hypertrophic–noncompaction phenotype.

The *MYL3* gene (MIM#160790) encodes myosin light chain 3 protein, belonging to the myosin family [5]. Myosins are essential in maintaining the structural integrity and shape of the cell. Moreover, the interaction with actins plays major role in myocyte contractility [6]. The alteration of the MYL3 protein (either hereditary or secondary due to the activation of caspase) results in the disruption of sarcomeres and eventually in contractile dysfunction due to the disorganized actin–myosin bonds in cardiomyocytes [7]. Pathogenic variants of the *MYL3* gene have been associated with familial HCM, type 8 (MIM#608751) inherited in autosomal dominant or autosomal recessive manner. To date, 41 different pathogenic variants have been identified in individuals affected with HCM [8]. The alteration NM_000258.2c.382G>T, NP_000249.1:p.(Gly128Cys), rs199474704 has been reported in patients with end-stage HCM [9]. The variant leads to amino acid change in a conservative position (EF–hand domain/EF–hand domain pair domain of the protein), and the biochemical difference between glycine and cysteine is high (Grantham score 159). In silico analysis results: SIFT—deleterious (score 0.01), PolyPhen2—probably damaging (score 0.927), Mutation taster—disease causing (score 0.99). The variant is listed in The Human Mutation Database (CM117919), and the frequencies in the 1000 Genomes Project and the ExAC project are 0.0 and 0.0, respectively. Functional analyses have not been performed. Based on these observations, the alteration was classified as possibly pathogenic according to the ACMG criteria, and segregation analysis in the family was recommended.

Another heterozygous missense type *MYH6* gene variant (NM_002471.3:c.169G>A, NP_002462.2:p.(Gly57Ser)) was also identified in the patient. The *MYH6* gene (MIM#160710) encodes an alpha heavy chain myosin, functioning as a fast ATPase in cardiac muscle and participating in the contraction of myocytes [10]. The alteration of alpha-MHC protein may lead to the switch of expression to the *MYH7* gene located upstream of the *MYH6* gene, thus accelerating the development of HCM [11]. Pathogenic variants of

the *MYH6* gene have been associated with atrial septal defect type 3 (MIM#614089), dilated CM type 1EE (MIM#613252), HCM type 14 (MIM#613251), and sick sinus syndrome type 3 (MIM#614090) [12]. The alteration in *MYH6* gene NM_002471.3:c.169G>A, NP_002462.2:p.(Gly57Ser) has not been described in scientific literature previously. The variant causes amino acid change in a semiconservative position (SH3-like/P-loop containing nucleoside triphosphate hydrolase domain at the N-terminal of myosin), but the biochemical difference between glycine and serine is moderate (Grantham score 56). In silico analysis results: SIFT—deleterious (score 0.024), PolyPhen2—benign (score 0.256), Mutation taster—disease causing (score 0.99). The frequency of the variant in the 1000 Genomes Project is 0.0; ExAC project—0.0. Functional analyses have not been performed. The genetic change was classified as a variant of unknown significance (VUS) according to the ACMG criteria.

The segregation analysis performed on the daughter of the patient revealed the same variants of *MYL3* and *MYH6* genes as her father. These findings uncovered the nature of the variants, making them clinically important. However, we currently are not able to discriminate how much they would influence disease development if they acted separately, and at which age we will see clear phenotypical CM expression. Nevertheless, the possession of two very rare alterations in CM genes might be associated with faster development of the disorder and with the diverse manifestation of structural rearrangements (both HCM and LVNC are present in our patient).

The overlapping phenotype in the adult patient described was diagnosed based on the established diagnostic criteria of both CM. The best imaging modality to prove phenotypical features is definitely CMR imaging. In our adult patient, HCM was diagnosed based on the European Society of Cardiology (ESC) guidelines for the diagnosis and management of HCM. In the latter guidelines, HCM is defined by a wall thickness ≥15 mm in one or more LV myocardial segments—as measured by any imaging technique (echocardiography, CMR or CT)—which is not explained solely by loading conditions [2]. In our patient, apical segments proximal to the apical microaneurysm were measured to be up to 16 mm in diastole, which is consistent with diagnostic criteria for HCM (Figure 1A,B,E,F). Additionally, next to the hypertrophied compact part of LV, we noticed a noncompaction area, which was more prominent at the level of midventricular segments. In order to diagnose noncompaction, we used the CMR criteria proposed by Petersen et al. [13]. Using the latter criteria, a diastolic non-compacted to compacted ratio >2.3 identifies pathological noncompaction with values for sensitivity, specificity, and positive and negative predictions of 86%, 99%, 75%, and 99%, respectively. Thus, a diastolic non-compacted to compacted myocardial ratio equal to 2.8 was consistent with the diagnosis of left ventricular noncompaction at the level of midventricular segments (Figure 1C,D). Taking into account two phenotypical features in our patient, we considered our patient to have an overlapping hypertrophic and noncompaction phenotype.

From a clinical point of view, clinicians should take into account the presence of red flags (RFs) known to be associated with specific systemic disease in a patient with HCM and other CMs. In our patient, we found a couple of features consistent with a non-syndromic type of CM. First of all, the patient's daughter has the same variants of MYL3 and MYH6 as her father. Additionally, the proband has multiple individuals affected with unspecified cardiac disorder in his family. Other clues were LV hypertrophy, repolarization abnormalities on the ECG, and LV apical hypertrophy with apical microaneurysm on cardiac imaging. We did not find any RF associated with non-sarcomeric (syndromic, skeletal myopathy, or infiltrative phenotype) HCM. According to the literature, the presence of RFs shows a high negative predictive value to exclude any specific (non-sarcomeric) HCM disease (98% [95%CI 94–99%]) [14].

The choice of the best treatment and surveillance strategy in patients with CM requires a personalized, specific, and experienced approach. Therefore, we involved a multi-disciplinary dedicated cardiomyopathy team composed of a clinical cardiologist, an interventional cardiologist, a cardiac surgeon, and an electrophysiologist in the decision-

making regarding the further treatment strategy for our patient. The strategic role of cardiomyopathy teams working in experienced centers has been suggested as a critical step in the management of cardiomyopathy patients [15].

There are currently no trials or predictive models to guide ICD insertion specifically for apical HCM or overlapping cardiomyopathy phenotypes. The ESC five-year HCM sudden cardiac death (SCD) risk score [16,17] was based on all HCM morphological subtypes without breakdown for apical HCM [16]. Maron's group has proposed an enhanced American College of Cardiology/American Heart Association (ACC/AHA) guideline-based risk factor algorithm strategy for HCM patients fulfilling one or more major risk factors for SCD. This novel risk prediction strategy includes novel high-risk markers, such as CMR LGE demonstration of extensive fibrosis comprising ≥15% of LV mass by quantification or "extensive and diffuse" by visual estimation, as well as the presence of LV apical aneurysm, independent of size, with associated regional scarring [18]. Compared with enhanced ACC/AHA risk factors, the ESC risk score [16,17] retrospectively applied to the study patients was much less sensitive than the ACC/AHA criteria (34% [95% CI, 22–44] vs. 95% [95% CI, 89–99]), consistent with recognizing fewer high-risk patients. Additionally, the main indications for ICD implantation for the primary prevention of sudden cardiac death that may apply to patients with LV noncompaction are patients with CM and LV ejection fraction ≤35% and high-risk HCM with LVNC [19]. Thus, we took into account the history of sudden cardiac death in the first-degree relative under 40 years of age (the proband's sister) and the presence of LV apical micro-aneurysm with associated transmural LGE (a major risk marker) to justify the recommendation for prophylactic ICDs in our patient.

## 5. Conclusions

In summary, our case suggests that the phenotypical expression of both LVNC and HCM can occur in the same patient having two different alterations (one of them likely pathogenic and the second is a VUS so far). This case also highlights the need for comprehensive multimodality imaging for simultaneous morphological and functional evaluation of the heart in order to assess the correct phenotype and to stratify the patient's risk. In the clinical routine, awareness of the existence of complex cardiomyopathy phenotypes should be raised during echocardiographic examination and should encourage the broader use of CMR.

**Supplementary Materials:** The following are available online at https://www.mdpi.com/2035-8 148/11/1/5/s1, Video S1: Magnetic resonance four chamber cine heart view. Video S2: Magnetic resonance two chamber cine heart view. Video S3: Magnetic resonance three chamber cine heart view. Video S4: Magnetic resonance midventricular short axis cine heart view. Video S5: Magnetic resonance apical short axis cine heart view. Video S6: Magnetic resonance two chamber heart cine view of patient's daughter: Video S7: Magnetic resonance apical short axis cine view of patient's daughter. Figure S1: Genealogy of the family (black symbols denote patients with detected variants in MYL3 and MYH6 genes. Half-filled symbol in II-3 denotes congenital heart defect. I-1 and 1–2 family members were affected with not specified cardiac disorders). Figure S2: The electropherogram of Sanger sequencing of MYL3 gene variant NM_000258.2:c.382G>T, NP_000249.1:p.(Gly128Cys), rs199474704 (reverse strand). Figure S3: The electropherogram of Sanger sequencing of MYH6 gene variant NM_002471.3:c.169G>A, NP_002462.2:p.(Gly57Ser) (forward strand).

**Author Contributions:** Conceptualization, S.G. and N.R.V.; writing—original draft preparation, S.G., E.P. and V.M.; writing—review and editing, S.G., V.M., E.P., R.N., R.J., and N.R.V.; visualization, S.G.; supervision, S.G. and N.R.V. All authors have read and agreed to the published version of the manuscript.

**Funding:** This research received no external funding.

**Institutional Review Board Statement:** Not applicable.

**Informed Consent Statement:** Informed consent was obtained from the patient.

**Data Availability Statement:** Password protected data supporting reported results can be found at https://eli.santa.lt/ (accessed on 4 August 2019) (could be provided anonymized data upon request by authors).

**Conflicts of Interest:** The authors declare no conflict of interest.

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
