# Peer review of "Overlapping Phenotype of Cardiomyopathy in a Patient with Double Mutation: A Case Report"

_cardiogenetics, doi:10.3390/cardiogenetics11010005_

Round 1

Reviewer 1 Report

In this manuscript, the authors investigated the new mutations in MYL3 and MYH6 genes that encode for the sarcomere proteins (MYL3 and MYH6 protein in the human respectively ), which could be beneficial for monitoring Left ventricular noncompaction (LVNC) and hypertrophic cardiomyopathy (HCM) disease progression.

They also investigated the morphology of the heart chambers using Cine MRI and suggested that these mutations lead to both diseases (LVNC and HCM) in the same patient.

The data is convincing, but few minor concerns need to be addressed for further move.

Please consider the below-mentioned points regarding this manuscript:

  1. Please expand the introduction for a better understanding of non-specialist readers.
  2. Please write the methods in detail as a separate paragraph for clear understanding.
  3. The Authors should acknowledge the patient and his family for allowing them to write this case report.

Author Response

  1. Please expand the introduction for a better understanding of non-specialist readers.

Authors: the introduction section has been expanded:

“Hereditary cardiomyopathies (CMs) represent a very large and heterogeneous group of inherited heart disorders. According to the results of the instrumental evaluation, many types of the disease have been characterized: hypertrophic CM (HCM), left ventricular noncompaction (LVNC), dilated CM, and arrhythmogenic right ventricle CM to mention the most frequent entities. LVNC (ORPHA:54260) and HCM (ORPHA:217569) commonly occur as separate disorders with distinct clinical and pathoanatomical features [1, 2]. However, in some patients, overlapping or mixed phenotypes are diagnosed only based on the use of sophisticated imaging modalities, especially cardiovascular magnetic resonance (CMR). Unfortunately, sometimes the phenotypic manifestation of overlapping phenotypes poses some difficulties in determination of the particular disorder that is essential in the treatment and surveillance of the patient. Moreover, the same genes may be implicated in pathogenesis of different CMs, making diagnostics even more complicated. Pathogenic variants in MYBPC3 and MYH7 genes are responsible for the development of most non-syndromic HCMs and a big part of LVNC cases, although more than 30 genes are currently associated with pathogenesis of these disorders.

In this case report, authors present two patients from one nuclear family suffering with hereditary heart disorders possessing variants in the MYL3 gene (MIM#160790) and the MYH6 gene (MIM#160710). Though we know that the pathogenic variants of the MYL3 gene have been associated with familial HCM, the alteration in the MYH6 gene NM_002471.3:c.169G>A, NP_002462.2:p.(Gly57Ser) has not been described in the scientific literature previously. The genetic change was classified as variant of unknown significance (VUS) according to the American College of Medical Genetics and Genomics (ACMG) criteria. Based on the case presented, we postulate that the presence of this double mutation causes an overlapping hypertrophic-noncompaction phenotype. Additionally, we emphasize the utility of multimodality imaging for the phenotypical assessment of cardiomyopathy patients, as well as a cardiomyopathy team approach to choose the most suitable treatment.”

  1. Please write the methods in detail as a separate paragraph for clear understanding.

Authors: the methods section has been introduced:

     “Next-generation sequencing. Next-generation sequencing analysis of genomic DNA isolated from two patients’ peripheral blood was performed using TruSight Cardio Sequencing panel (Illumina Inc., San Diego, California, USA). A total of 174 genes (coding exons) were analyzed, including the main genes associated with cardiomyopathies (ABCC9, ACTC1, ACTN2, ANKRD1, BRAF, CAV3, CBL, CRYAB, CSRP3, DES, DSC2, DSG2, DSP, DTNA, GAA, GLA, HFE, HRAS, JUP, KRAS, LAMA4, LAMP2, LDB3, LMNA, MAP2K1, MAP2K2, MYBPC3, MYH6, MYH7, MYL2, MYL3, MYLK2, MYOZ2, MYPN, NEXN, NRAS, PKP2, PLN, PRDM16, PRKAG2, PTPN11, RAF1, RBM20, RYR2, SCN5A, SGCD, SHOC2, SOS1, TAZ, TCAP, TGFB3, TMEM43, TNNC1, TNNI3, TNNT2, TPM1, TTN, TTR, VCL). Prepared DNA libraries were sequenced on the Illumina MiSeq system. The combined coverage was 572 kbp in sequence length. Data analysis was performed using standard Illumina bioinformatic workflow. Detected gene variants were analyzed and annotated using the VariantStudio 3.0 software. Synonymous or intronic variants and variants with a minor allele frequency of less than 2% were excluded. In silico analysis of missense mutations was performed using PolyPhen-2 (http://genetics.bwh.harvard.edu/pph2/), SIFT Human Protein (http://sift.jcvi.org/), and Mutation Taster (www.mutationtaster.org/).

     Sanger sequencing. Polymerase chain reactions (PCR) of gDNA sequences flanking variant NM_000258.2:c.382G>T, NP_000249.1:p.(Gly128Cys), rs199474704 of the MYL3 gene and variant NM_002471.3:c.169G>A, NP_002462.2:p.(Gly57Ser) of the MYH6 gene were performed using specific primers designed with the Primer Blast tool [3, 4]. The PCR products were sequenced using BigDye® Terminator v3.1 Cycle Sequencing Kit (Thermo Fisher Scientific, USA) and the ABI 3130xL Genetic Analyzer (Thermo Fisher Scientific, USA). The sequences were aligned with the reference sequence of the MYL3 (NCBI: NM_000258.2) and MYH6 (NCBI: NM_002471.3) genes.

     Imaging equipment. Two-dimensional transthoracic echocardiography (TTE) using an ultrasonic system equipped with 1.5-4.5 MHz transducer (GE Vivid E9, GE Healthcare, New York, USA) and 1.5 T cardiovascular magnetic resonance (CMR) (Siemens Avanto, Erlangen, Germany) to assess the specific CM phenotype.”

  1. The Authors should acknowledge the patient and his family for allowing them to write this case report.

Authors: please find corrections below (patients informed consent has been attached)

“Informed Consent Statement: Informed consent was obtained from the patient.

Acknowledgments: Informed consent was obtained from the patient.”

Reviewer 2 Report

Dear editor:

Thank you for inviting me to evaluate this article titled “Overlapping phenotype of cardiomyopathy in patient with double mutation”. This manuscript reports a patient with cardiomyopathy. The text is not well arranged and the logic is not clear. There is a lack of introduction of their experimental designs and data analysis used in the study (see comment 1 in major comments). Some of their conclusions are not convincing (see comment 2 and 3 in major comments). So I recommend to you that this manuscript cannot be accepted. The following are the major and minor questions in this manuscript:

Major comments:

  1. Line 52: “Patient underwent genetic consultation and testing. The latter revealed a heterozygous missense…” Genetic testing is an important part of this study and should be clearly introduced. If the authors use next generation sequencing to perform genetic testing, crucial work such as wet lab experimental designs (whole-genome sequencing, whole-exome sequencing, or targeted region sequencing) and bioinformatic analysis (software, sequencing data filtering, variant calling and filtering) should be described. Unfortunately, the authors did not present any information in the manuscript.
  2. The authors should validate the two candidate variants (Sanger sequencing for instance), and present the validated results as the evidence to solidify their genetic results.
  3. As a genetic disease study, the authors need to provide a detailed family history of the patient to reinforce their conclusions. Since the patient has two rare mutations, the probability that both rare mutations are de novo in the patient is extremely low, and there is a high probability that one or two mutations are inherited from the patient's parents. In addition, the authors indicated that the patient has a daughter. The candidate mutations and similar phenotypes need to be confirmed from the patient’s daughter.
  4. Line 18: The authors listed “next generation sequencing” as key words, but there is no description in the entire manuscript.

Minor comments:

  1. Line 88: “The alteration NM_000258.2c.382G>T, NP_000249.1:p.(Gly128Cys), rs199474704 has been reported in patient with end-stage HCM”. The authors should cite references here to support their conclusions.
  2. The authors listed the same RefSNP ID (rs number) of the two variants in the manuscript, but the rs number of the variant in MYH6 gene is wrong.

Author Response

Major comments:

  1. Line 52: “Patient underwent genetic consultation and testing. The latter revealed a heterozygous missense…” Genetic testing is an important part of this study and should be clearly introduced. If the authors use next-generation sequencing to perform genetic testing, crucial work such as wet-lab experimental designs (whole-genome sequencing, whole-exome sequencing, or targeted region sequencing) and bioinformatic analysis (software, sequencing data filtering, variant calling, and filtering) should be described. Unfortunately, the authors did not present any information in the manuscript.

Authors: We have added phenotypic evaluation of the patient and genealogy analysis to the manuscript. Also, a thorough description of new generation sequencing of targeted genes implicated in HCM development was provided. We added the following information in the Methods section:

“Next-generation sequencing analysis of genomic DNA isolated from patient‘s peripheral blood was performed using TruSight Cardio Sequencing panel (Illumina Inc., San Diego, California, USA). A total of 174 genes (coding exons) were analyzed, including the main genes associated with cardiomyopathies (ABCC9, ACTC1, ACTN2, ANKRD1, BRAF, CAV3, CBL, CRYAB, CSRP3, DES, DSC2, DSG2, DSP, DTNA, GAA, GLA, HFE, HRAS, JUP, KRAS, LAMA4, LAMP2, LDB3, LMNA, MAP2K1, MAP2K2, MYBPC3, MYH6, MYH7, MYL2, MYL3, MYLK2, MYOZ2, MYPN, NEXN, NRAS, PKP2, PLN, PRDM16, PRKAG2, PTPN11, RAF1, RBM20, RYR2, SCN5A, SGCD, SHOC2, SOS1, TAZ, TCAP, TGFB3, TMEM43, TNNC1, TNNI3, TNNT2, TPM1, TTN, TTR, VCL). Prepared DNA libraries were sequenced on the Illumina MiSeq system. The combined coverage was 572 kbp in sequence length. Data analysis was performed using standard Illumina bioinformatics workflow. Detected gene variants were analyzed and annotated using VariantStudio 3.0 software. Synonymous or intronic variants and variants with a minor allele frequency >2% have been excluded. In silico analysis of missense mutations was performed using PolyPhen-2 (http://genetics.bwh.harvard.edu/pph2/), SIFT Human Protein (http://sift.jcvi.org/) and Mutation Taster (www.mutationtaster.org/).”

  1. The authors should validate the two candidate variants (Sanger sequencing for instance), and present the validated results as evidence to solidify their genetic results.

Authors: Sanger sequencing of the variants was performed on the patient’s daughter for the analysis of disease segregation. The identification of the genetic changes in her DNA sample also served as a validation of the results of new generation sequencing of the patient. This information we added to the manuscript in sections Methods and Results.

  1. As a genetic disease study, the authors need to provide a detailed family history of the patient to reinforce their conclusions. Since the patient has two rare mutations, the probability that both rare mutations are de novo in the patient is extremely low, and there is a high probability that one or two mutations are inherited from the patient's parents. In addition, the authors indicated that the patient has a daughter. The candidate mutations and similar phenotypes need to be confirmed from the patient’s daughter.

Authors: Detailed family history is provided, information about results of segregation analysis presented. Unfortunately, the parents and sister of the patient have already diseased and other more distant family members were not available for genetic testing. We provided the information about parent’s health condition in the section Case report:

“The parents of the patient could not be tested. The father died of myocardium infarction at age of 72 years, the mother had heart rhythm disorder, and died at age of 64 years. The sister of the patient experienced sudden cardiac death at the age of 5 has not specified a “congenital” heart defect.”

We also included clinical information about the daughter of the patient, who is affected too.

  1. Line 18: The authors listed “next generation sequencing” as key words, but there is no description in the entire manuscript.

Authors: Thank you for this remark, we added a description of new generation sequencing of targeted genes, as mentioned in previous comments.

 Minor comments:

  1. Line 88: “The alteration NM_000258.2c.382G>T, NP_000249.1:p.(Gly128Cys), rs199474704 has been reported in patient with end-stage HCM”. The authors should cite references here to support their conclusions.

Authors: The reference to support the statement regarding the previous publication of the variant was added.

  1. The authors listed the same RefSNP ID (rs number) of the two variants in the manuscript, but the rs number of the variant in MYH6 gene is wrong.

Authors: The listing the same RefSNP ID (rs number) of the two variants in the manuscript was the “lapsus scriptum”. The rs199474704 should be attributed only to MYL3 gene variant. We have deleted it where it was not appropriate and apologize for the mistake. MYH6 gene variant identified to the patient does not have rs number yet.

Reviewer 3 Report

In the present manuscript, the authors presented a clinical case of a 39-year-old man with genetic mutations, causing a dual clinical phenotype of LVNC and HCM. The case is of interest and well presented.

Major comments: None.

Minor comments:

  • I suggest revising the manuscript in compliance with the CARE reporting guidelines. These guidelines, for example, suggest to report “case report” in the title. A CARE checklist should also be added as supplemental materials.
  • It worth notice that the present case was discussed by a dedicated cardiomyopathy team. This concept has been recently proposed as the optimal approach in patients with genetic cardiac disease (Pelliccia et al. Int J Cardiol. 2020;304:86-92). I suggest underlining and better discuss this relevant concept in the text.
  • What are the “red flags” that should guide clinicians’ decisions in this particular subset of patients? Please discuss (Limongelli et al. Int J Cardiol. 2020;299:186-191).
  • Please, check the manuscript for typos and grammatical errors, and improve the English presentation. For example, instead of "...which was diagnosed by transthoracic echocardiography (TTE) 3 years ago.", I would say "...3 years before".

Author Response

Major comments: None.

Minor comments:

  • I suggest revising the manuscript in compliance with the CARE reporting guidelines. These guidelines, for example, suggest to report “case report” in the title. A CARE checklist should also be added as supplemental materials.

Authors: Thank you for your suggestion. The title (and other parts of the manuscript) was corrected according to CARE reporting guidelines:

“Overlapping phenotype of cardiomyopathy in a patient with double mutation: case report.”

-Keywords has been corrected with the inclusion of “case report”:

“Keywords: left ventricular noncompaction, apical hypertrophic cardiomyopathy, next-generation sequencing, case report”.

-A CARE checklist is attached (please see attachment).

  • It worth notice that the present case was discussed by a dedicated cardiomyopathy team. This concept has been recently proposed as the optimal approach in patients with genetic cardiac disease (Pelliccia et al. Int J Cardiol. 2020;304:86-92). I suggest underlining and better discuss this relevant concept in the text.

Authors: The fact that our adult patient was discussed in a dedicated cardiomyopathy team has been highlighted in the case described previously. I the discussion section we included the reason to have a dedicated cardiomyopathy team and its composition in our center:

“The choice of the best treatment and surveillance strategy in patients with cardiomyopathies needs personalized, specific and experienced approach. Therefore, we involved a multidisciplinary dedicated cardiomyopathy team composed of a clinical cardiologist, an interventional cardiologist, a cardiac surgeon, and an electrophysiologist in the decision-making regarding the further treatment strategy of our patient. The strategic role of cardiomyopathy teams working in experienced centers has been suggested recently as a critical step in the management of cardiomyopathy patients (Pelliccia et al. Int J Cardiol. 2020;304:86-92)”.

  • What are the “red flags” that should guide clinicians’ decisions in this particular subset of patients? Please discuss (Limongelli et al. Int J Cardiol. 2020;299:186-191).

Authors: The “red flags” approach has been described in the discussion section:

“From a clinical point of view clinicians should take into account the presence of red flags (RF), known to be associated with specific systemic disease in a patient with HCM and other cardiomyopathies. In our patient, we found a couple of features consistent with the non-syndromic type of HCM. First of all, the patient’s daughter has the same variants of MYL3 and MYH6 genes inherited from her father. Additionally, proband has multiple individuals affected with the unspecified cardiac disorder in his family. Other clues were left ventricular hypertrophy and repolarization abnormalities on ECG, LV apical hypertrophy with apical microaneurysm on cardiac imaging. We didn’t find any RF associated with non-sarcomeric (syndromic, skeletal myopathy, or infiltrative phenotype) HCM. According to the literature, the presence of RF shows a high negative predictive value to exclude any specific (non-sarcomeric) HCM disease (98% [95%CI 94–99%]) (Limongelli et al. Int J Cardiol. 2020;299:186-191).

  • Please, check the manuscript for typos and grammatical errors, and improve the English presentation. For example, instead of "...which was diagnosed by transthoracic echocardiography (TTE) 3 years ago.", I would say "...3 years before".

Authors: Thank you for this suggestion. The manuscript has been checked by the professional English language editor with a specialization in medical texts. The edited manuscript is submitted separately.

Reviewer 4 Report

The authors have identified an interesting case and have valuable phenotypic and genotypic data to share.

There are wording issues with this report but more pressing concerns as well. I do not understand the patient’s presenting concern or pertinent negative symptoms such as exercise capacity, presyncope, etc.

The authors do not clearly define the features that enabled them to diagnose apical variant HCM or LVNC and did not provide evidence-based criteria to convince the reader these conditions were present and to what severity.

I am not certain I understand the rationale for an ICD—was this entirely patient preference or were other criteria used to make this decision.

Was anticoagulation considered? 

Seems like this patient has multiple mutations that have been documented in HCM or at least myosin related. I don't see anything associated to non-compaction.

Do the authors think there is a genetic explanation for the LVNC?

The authors didn’t do a great job providing a genetic explanation for this unusual case or postulating a novel theory about why these two conditions are coexisting.

Spell checks

line 21 “ant”

Figure legend “resonance”

Author Response

There are wording issues with this report but more pressing concerns as well. I do not understand the patient’s presenting concern or pertinent negative symptoms such as exercise capacity, presyncope, etc.

Authors: Thank you for the important point. The rationale of the case report is to present mixed phenotypical cardiomyopathy patterns with double mutations. However, we know that the pathogenic variants of MYL3 gene have been associated with familial hypertrophic cardiomyopathy, the alteration in MYH6 gene NM_002471.3:c.169G>A, NP_002462.2:p.(Gly57Ser) has not been described in scientific literature previously. The variant causes amino acid change in semiconservative position (myosin, N-terminal, SH3-like/ P-loop containing nucleoside triphosphate hydrolase domain of the protein) but the biochemical difference between glycine and serine is moderate (Grantham score 56). In silico analysis results: SIFT – deleterious (score 0.024), PolyPhen2 – benign (score 0.256), Mutation taster – disease-causing (score 0.99). The frequency of the variant in the 1000 genomes project is 0.0, ExAC project - 0.0. Functional analyses have not been performed. The genetic change was classified as variant of unknown significance (VUS) according to the ACMG criteria. Thus, we think that the presence of this double mutation causes overlapping hypertrophic – non-compaction phenotype.

The authors do not clearly define the features that enabled them to diagnose apical variant HCM or LVNC and did not provide evidence-based criteria to convince the reader these conditions were present and to what severity.

Authors: Please find the discussion in the discussion section.

“The overlapping phenotype in the adult patient described was diagnosed based on the established diagnostic criteria of both CM. The best imaging modality to prove pheno-typical features is definitely CMR imaging. In our adult patient, HCM was diagnosed based on the European Society of Cardiology (ESC) guidelines for the diagnosis and management of HCM. In the latter guidelines, HCM is defined by a wall thickness ≥15 mm in one or more LV myocardial segments—as measured by any imaging technique (echocardiography, CMR or CT)—that is not explained solely by loading conditions [13]. In our patient, apical segments proximal to the apical microaneurysm were measured to be up to 16 mm in diastole, which is consistent with diagnostic criteria for HCM (Figure 1A-B, E-F). Additionally, next to the hypertrophied compact part of LV, we noticed noncompaction area, which was more prominent at the level of midventricular segments. In order to diagnose noncompaction, we used the CMR criteria proposed by Petersen et al. [14]. Using the latter criteria, a diastolic non-compacted to compacted ratio >2.3 identifies pathological noncompaction with values for sensitivity, specificity, and positive and negative predictions of 86%, 99%, 75%, and 99%, respectively. Thus, a diastolic non-compacted to compacted myocardial ratio equal to 2.8 was consistent with the diagnosis of left ventricular noncompaction at the level of midventricular segments (Figure 1C-D). Taking into account two phenotypical features in our patient, we considered our patient to have an overlapping hypertrophic and noncompaction phenotype”.

I am not certain I understand the rationale for an ICD—was this entirely patient preference or were other criteria used to make this decision.

Authors: we explained our decision-making process in detail in the discussion section of the manuscript.

“There are currently no trials or predictive models to guide ICD insertion specifically for apical HCM or overlapping cardiomyopathy phenotypes. The ESC 5‐year HCM sudden cardiac death (SCD) risk score [17, 18] was based on all HCM morphological subtypes without breakdown for apical HCM [17]. Maron's group has recently proposed an enhanced American College of Cardiology/American Heart Association (ACC/AHA) guideline-based risk factor algorithm strategy for HCM patients fulfilling one or more major risk factors for SCD. This novel risk prediction strategy includes novel high‐risk markers, such as CMR LGE demonstration of extensive fibrosis comprising ≥15% of LV mass by quantification or “extensive and diffuse” by visual estimation and also the presence of LV apical aneurysm, independent of size, with associated regional scarring [19]. Compared with enhanced ACC/AHA risk factors, the ESC risk score [17, 18] retrospectively applied to the study patients was much less sensitive than the ACC/AHA criteria (34% [95% CI, 22-44] vs 95% [95% CI, 89-99]), consistent with recognizing fewer high-risk patients. Additionally, the main indications for ICD implantation for primary prevention of sudden cardiac death that may apply to patients with LV noncompaction are patients with CM and LV ejection fraction ≤35% and high-risk HCM with LVNC [20]. Thus, we took into account the history of sudden cardiac death in the first-degree relative under age of 40 (the proband’s sister) and the presence of LV apical micro-aneurysm with associated transmural LGE (a major risk marker) to justify recommendation for prophylactic ICDs in our patient”.

Was anticoagulation considered? 

Authors: Anticoagulation was not considered in our case, because anticoagulation on the basis of excessive trabeculation remains controversial (D’Silva, A., & Jensen, B. (2020). Heart, heartjnl–2020–316945. doi:10.1136/heartjnl-2020-316945). Our patient has no atrial fibrillation, hid CHADS2 score is equal to 0, he has no evidence of thrombi on cardiovascular magnetic resonance, no previous embolism, LV ejection fraction was in normal range. Thus, we think that anticoagulation in this case was very questionable and to suggest close follow-up was the best approach at the moment.

Seems like this patient has multiple mutations that have been documented in HCM or at least myosin related. I don't see anything associated to non-compaction.Do the authors think there is a genetic explanation for the LVNC?

Authors: We think that the alteration in MYH6 gene NM_002471.3:c.169G>A, NP_002462.2:p.(Gly57Ser) has not been described in scientific literature previously and the latter alteration can be responsible for overlapping phenotype.

The authors didn’t do a great job providing a genetic explanation for this unusual case or postulating a novel theory about why these two conditions are coexisting.

Authors: genetic part has been corrected and expanded. Please find captions from the manuscript text below:

In the introduction section:

“Hereditary cardiomyopathies (CMs) represent a very large and heterogeneous group of inherited heart disorders. According to the results of instrumental evaluation, many types of the disease have been characterized: hypertrophic CM (HCM), left ventricular noncompaction (LVNC), dilated CM, and arrhythmogenic right ventricle CM to mention the most frequent entities. LVNC (ORPHA:54260) and HCM (ORPHA:217569) commonly occur as separate disorders with distinct clinical and pathoanatomical features [1, 2]. However, in some patients, overlapping or mixed phenotypes are diagnosed only based on the use of sophisticated imaging modalities, especially cardiovascular magnetic resonance (CMR). Unfortunately, sometimes the phenotypic manifestation of overlapping phenotypes poses some difficulties in determination of the particular disorder that is essential in the treatment and surveillance of the patient. Moreover, the same genes may be implicated in pathogenesis of different CMs, making diagnostics even more complicated. Pathogenic variants in MYBPC3 and MYH7 genes are responsible for the development of most non-syndromic HCMs and a big part of LVNC cases, although more than 30 genes are currently associated with pathogenesis of these disorders.

In this case report, authors present two patients from one nuclear family suffering with hereditary heart disorders possessing variants in the MYL3 gene (MIM#160790) and the MYH6 gene (MIM#160710). Though we know that the pathogenic variants of the MYL3 gene have been associated with familial HCM, the alteration in the MYH6 gene NM_002471.3:c.169G>A, NP_002462.2:p.(Gly57Ser) has not been described in the scientific literature previously. The genetic change was classified as variant of unknown significance (VUS) according to the American College of Medical Genetics and Genomics (ACMG) criteria. Based on the case presented, we postulate that the presence of this double mutation causes an overlapping hypertrophic-noncompaction phenotype. Additionally, we emphasize the utility of multimodality imaging for the phenotypical assessment of cardiomyopathy patients, as well as a cardiomyopathy team approach to choose the most suitable treatment.”

In the case report description:

“Patient underwent genetic consultation and testing. Phenotypic evaluation revealed a non-syndromic type of HCM. Genealogy analysis showed multiple individuals affected with cardiac disorder in the family. The father died of myocardial infarction at the age of 72 years; the mother suffered from heart rhythm disorder and died at the age of 64 years. The sister of the patient experienced sudden cardiac death at the age of 5 having not specified “congenital” heart defect. The genetic testing of the patient revealed a heterozygous missense type MYL3 gene variant NM_000258.2c.382G>T, NP_000249.1:p.(Gly128Cys), rs199474704 and a heterozygous missense type MYH6 gene variant NM_002471.3:c.169G>A, NP_002462.2:p.(Gly57Ser).

Additionally, the patient’s 15-years-old daughter was invited for cardiological ex-amination and genetic consultation and testing. CMR revealed normal LV systolic function without evidence of LV hypertrophy. However, hypertrabeculation of apical to midventricular segments was observed with the ratio of non-compacted to compacted myocardium up to 2.0, which was not diagnostic for left ventricular noncompaction (see Supplementary videos 6-7). 24-hour ECG monitoring was performed and revealed one sporadic supraventricular premature beat. Sanger sequencing confirmed that the patient passed both MYL3 gene variant c.382G>T and MYH6 gene variant c.169G>A to his affect-ed daughter”.

In the methods section:

“Next-generation sequencing. Next-generation sequencing analysis of genomic DNA isolated from two patients’ peripheral blood was performed using TruSight Cardio Sequencing panel (Illumina Inc., San Diego, California, USA). A total of 174 genes (coding exons) were analyzed, including the main genes associated with cardiomyopathies (ABCC9, ACTC1, ACTN2, ANKRD1, BRAF, CAV3, CBL, CRYAB, CSRP3, DES, DSC2, DSG2, DSP, DTNA, GAA, GLA, HFE, HRAS, JUP, KRAS, LAMA4, LAMP2, LDB3, LMNA, MAP2K1, MAP2K2, MYBPC3, MYH6, MYH7, MYL2, MYL3, MYLK2, MYOZ2, MYPN, NEXN, NRAS, PKP2, PLN, PRDM16, PRKAG2, PTPN11, RAF1, RBM20, RYR2, SCN5A, SGCD, SHOC2, SOS1, TAZ, TCAP, TGFB3, TMEM43, TNNC1, TNNI3, TNNT2, TPM1, TTN, TTR, VCL). Prepared DNA libraries were sequenced on the Illumina MiSeq system. The combined coverage was 572 kbp in sequence length. Data analysis was performed using standard Illumina bioinformatic workflow. Detected gene variants were analyzed and annotated using the VariantStudio 3.0 software. Synonymous or intronic variants and variants with a minor allele frequency of less than 2% were excluded. In silico analysis of missense mutations was performed using PolyPhen-2 (http://genetics.bwh.harvard.edu/pph2/), SIFT Human Protein (http://sift.jcvi.org/), and Mutation Taster (www.mutationtaster.org/).

     Sanger sequencing. Polymerase chain reactions (PCR) of gDNA sequences flanking variant NM_000258.2:c.382G>T, NP_000249.1:p.(Gly128Cys), rs199474704 of the MYL3 gene and variant NM_002471.3:c.169G>A, NP_002462.2:p.(Gly57Ser) of the MYH6 gene were performed using specific primers designed with the Primer Blast tool [3, 4]. The PCR products were sequenced using BigDye® Terminator v3.1 Cycle Sequencing Kit (Thermo Fisher Scientific, USA) and the ABI 3130xL Genetic Analyzer (Thermo Fisher Scientific, USA). The sequences were aligned with the reference sequence of the MYL3 (NCBI: NM_000258.2) and MYH6 (NCBI: NM_002471.3) genes.”

In the discussion section:

“With this case report, authors present two patients from one nuclear family suffering from hereditary heart disorders possessing variants in the MYL3 gene (MIM#160790) and the MYH6 gene (MIM#160710). Based on the case description, we postulate that the presence of the particular double mutation causes an overlapping hypertrophic – noncompaction phenotype.

The MYL3 gene (MIM#160790) encodes myosin light chain 3 protein, belonging to the myosin family [5]. Myosins are essential in maintaining the structural integrity and shape of the cell. Moreover, the interaction with actins plays major role in myocyte contractility [6]. The alteration of the MYL3 protein (either hereditary or secondary due to the activation of caspase) results in disruption of sarcomeres and eventually in contractile dysfunction due to the disorganized actin-myosin bonds in cardiomyocytes [7]. Pathogenic variants of the MYL3 gene have been associated with familial HCM, type 8 (MIM#608751) inherited in autosomal dominant or autosomal recessive manner. To date, 41 different pathogenic variants have been identified in individuals affected with HCM [8]. The alteration NM_000258.2c.382G>T, NP_000249.1:p.(Gly128Cys), rs199474704 has been reported in patient with end-stage HCM [9]. The variant leads to amino acid change in a conservative position (EF-hand domain/EF-hand domain pair domain of the protein), and the biochemical difference between glycine and cysteine is high (Grantham score 159). In silico analysis results: SIFT – deleterious (score 0.01), PolyPhen2 – probably damaging (score 0.927), Mutation taster – disease causing (score 0.99). The variant is listed in The Human Mutation Database (CM117919), and the frequency in the 1000 Genomes Project and the ExAC project is 0.0 and 0.0, respectively. Functional analyses have not been performed. Based on these observations, the alteration was classified as possibly pathogenic according to the ACMG criteria, and segregation analysis in the family was recommended.

Another heterozygous missense type MYH6 gene variant (NM_002471.3:c.169G>A, NP_002462.2:p.(Gly57Ser)) was also identified to the patient. The MYH6 gene (MIM#160710) encodes an alpha heavy chain myosin, functioning as a fast ATPase in cardiac muscle and participating in the contraction of myocytes [10]. The alteration of alpha-MHC protein may lead to the switch of expression to the MYH7 gene located upstream of the MYH6 gene, thus accelerating the development of HCM [11]. Pathogenic variants of MYH6 gene have been associated with atrial septal defect type 3 (MIM#614089), dilated CM type 1EE (MIM#613252), HCM type 14 (MIM#613251), and sick sinus syndrome type 3 (MIM#614090) [12]. The alteration in MYH6 gene NM_002471.3:c.169G>A, NP_002462.2:p.(Gly57Ser) has not been described in scientific literature previously. The variant causes amino acid change in a semiconservative position (SH3-like/P-loop containing nucleoside triphosphate hydrolase domain at the N-terminal of myosin), but the biochemical difference between glycine and serine is moderate (Grantham score 56). In silico analysis results: SIFT – deleterious (score 0.024), Poly-Phen2 – benign (score 0.256), Mutation taster – disease-causing (score 0.99). The frequency of the variant in the 1000 Genomes Project is 0.0, ExAC project - 0.0. Functional analyses have not been performed. The genetic change was classified as variant of unknown significance (VUS) according to the ACMG criteria.

The segregation analysis performed to the daughter of the patient revealed the same variants of MYL3 and MYH6 genes as her father. These findings uncover the nature of the variants, making them clinically important. However, we currently are not able to discriminate how much they would influence disease development if acted separately and at which age we will see clear phenotypical CM expression. Nevertheless, the possession of two very rare alterations in CM genes might be associated with faster development of the disorder and with the diverse manifestation of structural rearrangements (both HCM and LVNC are present in our patient).”

Spell checks

line 21 “ant”

Figure legend “resonance”

Authors: the above-mentioned spell errors have been corrected.

Round 2

Reviewer 2 Report

Dear editor:

Most of the previous critiques have been addressed by the authors. However, there are still some issues are insufficiently addressed:

  1. Line 130: “Sanger sequencing confirmed that the patient passed both MYL3 gene variant c.382G>T and MYH6 gene variant c.169G>A to his affected 132 daughter.” The authors need to provide a figure here as a validation of the Sanger Sequencing results.
  2. The third issue of my last report: Providing a detailed family history of the patient can reinforce the authors’ conclusions. Specificity, a pedigree chart can intuitively display the family history of the patient. We suggest the authors add a pedigree chart to the manuscript as most previous genetic disease studies do.

Author Response

Most of the previous critiques have been addressed by the authors. However, there are still some issues are insufficiently addressed:

  1. Line 130: “Sanger sequencing confirmed that the patient passed both MYL3 gene variant c.382G>T and MYH6 gene variant c.169G>A to his affected 132 daughter.” The authors need to provide a figure here as a validation of the Sanger Sequencing results.

Authors: Thank you for this suggestion. Sanger sequencing results have been added as Supplementary figures 2 and 3. Reference in line 123.

Lines 281-284 (legend of the figure, highlighted in yellow).

  1. The third issue of my last report: Providing a detailed family history of the patient can reinforce the authors’ conclusions. Specificity, a pedigree chart can intuitively display the family history of the patient. We suggest the authors add a pedigree chart to the manuscript as most previous genetic disease studies do.

Authors: Thank you for this very good suggestion. We have added a pedigree chart as Supplementary figure 1.

Line 117 (a reference to the chart, highlighted in yellow)).

Lines 279-281 (legend of the figure, highlighted in yellow).

The figure has been attached

Reviewer 4 Report

The authors did a great job in explaining the case study.

Author Response

Authors: Thank you for your work as reviewer. We appreciate it very much.